# A Study of Accelerated Corrosion of Stainless Steels under Highly Oxidizing Conditions

Alberto Ubaldini [1,*], Chiara Telloli [1], Antonietta Rizzo [1], Alessandro Gessi [1], Giuseppe Marghella [1], Stefania Bruni [1], Sara Calistri [1], Francesco Gennerini [2] and Georgiana Pintilei [3]

[1] ENEA, Italian National Agency for New Technologies, Energy and Sustainable Economic Development, Via Martiri di Monte Sole 4, 40129 Bologna, Italy; chiara.telloli@enea.it (C.T.); antonietta.rizzo@enea.it (A.R.); alessandro.gessi@enea.it (A.G.); giuseppe.marghella@enea.it (G.M.); stefania.bruni@enea.it (S.B.); sara.calistri2@unibo.it (S.C.)

[2] Department of Electrical, Electronic and Information Engineering "Guglielmo Marconi" (DEI), Biomedical Engineering, Cesena Campus, Alma Mater Studiorum University of Bologna, Via dell'Università 50, 47522 Cesena, Italy

[3] Departement of Industrial Engineering, Alma Mater Studiorum University of Bologna, Via Saragozza 8, 40136 Bologna, Italy; georgiana.pintilei3@gmail.com

* Correspondence: alberto.ubaldini@enea.it

**Abstract:** The corrosion behavior of certain steels under extremely oxidative conditions, simulating the impact of water radiolysis on stainless steels, has been investigated. Radiolysis generates aggressive species, including radicals, solvated electrons, and hydrogen peroxide, potentially leading to corrosion over time in materials typically considered resistant. To expedite the kinetics of this phenomenon, drastic conditions were employed, involving high concentrations of peroxide in a strongly acidic environment. Under these conditions, corrosion can manifest rapidly. The varied responses of different steels are contingent upon their inherent nature and chemical composition, notably the chromium and nickel content. Steels with higher chromium and nickel concentrations exhibit increased resistance to corrosion, even in such severe environments. Microscopic corrosion mechanisms involve pitting and intergranular corrosion. Pitting results in the formation of craters on surfaces, while intergranular corrosion leads to the detachment of grains.

**Keywords:** corrosion; hydrogen peroxide; intergranular corrosion; pitting





## 1. Introduction

Stainless steels are among the most used materials in the world, in countless fields, due to the excellent combination of mechanical and chemical properties, cost, and durability. Among other properties, they are excellent for resistance to environmental corrosion [1,2], i.e., the irreversible deterioration of a material and of its properties due to the electrochemical or chemical reaction on its surface. More precisely, steel corrosion is an electrochemical process in which the metallic surface releases electrons into an electrolyte, such as a layer of moisture in the presence of oxygen [3]. Metallic oxides are generally more stable than metals themselves, and iron in the steel in a moist air environment slowly transforms into oxides as rust.

From a microscopic perspective, corrosion encompasses a broad spectrum of phenomena, each exhibiting unique characteristics [2]. Among the others, pitting, crevice corrosion, and intra- and intergranular corrosion should be cited [4]. The most effective way to protect steels and avoid corrosion is the control of the chemical composition by adding some specific elements.

Actually, stainless steels are alloys containing a minimum of about 12% Cr (in addition to other elements, present for other different purposes), which is the amount needed to prevent the formation of rust in unpolluted atmospheres; nickel in different amounts; and

a carbon content of 0.2 to 2.11%. They achieve their stainless characteristics through the formation of an invisible and adherent chromium-rich oxide film [5].

Stainless steels are very resistant in normal conditions, but in some aggressive situations, they can be subject to corrosion phenomena [3,6]. Among them, the marine environment, where steels are subjected to the corrosive effects of sodium chloride, must certainly be mentioned, but also, in certain cases, the physiological environment can be particularly harsh for materials of biomedical implants. Parts of mechanical components which are subjected to continuous and intense mechanical stress, such as bearings, or to high temperatures, such as turbines, or are in contact with various chemical substances, can be corroded much more easily. In all these cases, the conditions are very far from ideal, and the steels can deteriorate quickly. This aspect is very important for some applications, for instance, their use in nuclear plants or in particle accelerator apparatuses [7].

Stainless steels are very widely used in the nuclear industry, in particular, the austenitic 300 series, which represents more than a third of the entire stainless steel production. The two most widespread types of stainless steel are the AISI 304 type, which is considered as the standard grade, widely used in applications requiring a good combination of corrosion resistance and formability [2], and the AISI types 316 and 321, which have improved corrosion resistance through the addition of molybdenum. Whereas the first one has been much investigated, and in detail [8–10], other types of steels are somewhat less studied; therefore, this work focuses more on them.

The commissioning of nuclear power reactors requires materials that meet strict criteria regarding their mechanical properties, thermal and chemical stability, and corrosion resistance [11]. This is because radiation can cause crystalline structural stresses in materials, which can result in significant changes to their properties compared to normal conditions. Therefore, only materials that can withstand such conditions are used in nuclear reactors. In such a case, even steel reactivity in regard to water can be very different from what is observed in more standard conditions, because of operating conditions in terms of temperature, pressure, and external parameters, and because of intrinsic physical phenomena, in particular, the water radiolysis. Very briefly, water radiolysis is the dissociation of water molecules due to radiation, directly in the case of ionizing radiation or indirectly in the case of high-energy neutron beams [12,13]. As a consequence of radiolysis, highly reactive species, such as radicals, free electrons, ions, and atomic oxygen form, and the pH of the system can be far from neutrality [13].

Some of these species, such as hydrated electrons and hydrogen atoms, are very strong reducing agents, whereas hydroxyl radicals are very strong oxidative ones and, in their presence, even resistant steels can undergo much more severe corrosion than what could be expected during a typical use, reducing the lifetime of certain components. Many operational and safety issues of nuclear plants for energy production stem from the effects of radiation on the oxidation or corrosion kinetics of construction materials. The responses of steels subjected to the action of water after radiolysis can be very different from those under standard conditions [14–16]. This is an aspect to take into consideration when designing the systems, and it is, in any case, interesting to study the behavior of materials in these conditions. It is also interesting to observe that high corrosion rates ($C_R$) can be observed in steels, even stainless steels, when they are in contact with saline solutions, i.e., in the presence of a high concentration of inorganic cations and anions, such as alkaline ions, halides, nitrates, sulfides, sulfates, etc., but radiolysis can create an aggressive media even in their absence.

Under standard conditions, the corrosion rate for a typical steel is only a few microns per year or less, which means that these processes are still very slow and can take years to complete. Furthermore, it is difficult to monitor and control radiolysis at the laboratory scale in a safe and secure way [17,18], all for a very long time, making it difficult to study the kinetics and microscopic mechanisms of the corrosion.

In order to overcome these problems and to perform this kind of study within a reasonable time, it would necessary to emulate the corrosion process in an accelerated

manner, increasing the concentration of the aggressive species. Thus, it would be possible to study the kinetics and infer the long-term behavior.

One potential method to accomplish this is by utilizing solutions with high concentrations of hydrogen peroxide, which happens to be one of the byproducts of water radiolysis, at varying temperatures and high pH levels. In this way, materials can be subjected to very oxidative conditions, and it is possible to observe their long-term behavior over shorter ones, and it is possible to carry out an accelerated study of their corrosion.

The aim of the present work is therefore to investigate the behavior of different stainless steels in the presence of concentrated hydrogen peroxide solutions, at a high temperatures and different pHs.

Sample evolution and kinetics have been studied by measuring the mass variation as a function of time under different conditions, while SEM-EDS has been used to determine changes in microstructures resulting from corrosion. The presence of small amounts of secondary phases has been analyzed by Raman spectroscopy.

## 2. Materials and Methods

Different samples of stainless steels, with masses on the order of a few hundred milligrams, were put in contact with commercial hydrogen peroxide, in large excess, changing the experimental conditions. These samples were cut into small, roughly parallelepiped, pieces of a few mm on each side, in order to have comparable exposed surfaces. Table 1 shows the steels considered and their nominal compositions. The average nominal compositions are deduced from the technical data sheets of the main international steel producers. However, EDX analyses carried out on the samples were found to be in excellent agreement with them. In the continuation, only the compositions indicated will be taken into consideration.

**Table 1.** Chemical composition of stainless steels.

| Sample | AISI Grade | Element Percentages [%] | | | | | | | | | |
|---|---|---|---|---|---|---|---|---|---|---|---|
| | | Cr | Ni | C | Si | Mn | P | S | Mo | N | Ti |
| 1 | 316 | 16–18 | 10–14 | 0.08 | 0.75 | 2 | 0.045 | 0.03 | 2–3 | 0.1 | |
| 2 | 316L | 16.5–18.5 | 10–13 | 0.03 | 1 | 2 | 0.045 | 0.015 | 2–2.5 | 0.1 | |
| 3 | 316Ti | 16.5–18 | 10.50–13.5 | 0.08 | 1 | 2 | 0.045 | 0.015 | 2–2.5 | | 0.2–0.7 |
| 4 | 321 | 17–19 | 9–12 | <0.08 | <1 | <2 | <0.045 | <0.015 | 0–0.3 | | 0.15–0.7 |
| 5 | 321H | 17–19 | 9–12 | 0.1 | 1 | 2 | 0.045 | 0.015 | 0–0.3 | 0.5–08 | |
| 6 | 347 | 17–19 | 9–12 | 0.08 | 1 | 2 | 0.045 | 0.015 | 0–0.25 | | |

Hydrogen peroxide concentration ranged from 10% to 40% $m/v$ and temperature from 5 °C to 80 °C, whereas the time of the experiments varied between 30 and 480 min. To modify its electrochemical potential, acidic solutions were prepared by adding 1%, 5%, and 10% by volume of hydrochloric, 1 M or concentrated (35% $m/v$), acid to the hydrogen peroxide solution. In order to avoid self-decomposition, hydrogen peroxide was stored in a refrigerator below −10 °C.

The morphology and the structure of the samples after the treatments were analyzed using an FEI Quanta Inspect S Scanning Electron Microscope coupled to an Oxford Instruments Xplore microanalyzer. Characterization of the samples, morphology and composition has been carried out at different magnifications using both secondary electrons and backscattered electron detectors at a 10 mm working distance, with energy ranging from 10 to 20 kV. The elemental analysis was carried out in the most significant areas of the samples. Sample surfaces were also investigated by Raman spectroscopy, acquiring their spectra at room temperature, with a BWTEK i-Raman plus spectrometer equipped with a 785 nm laser in the range 100–3500 cm$^{-1}$ with a spectral resolution of 2 cm$^{-1}$, fixing the measurement parameters such as acquisition time, number of repetitions, and laser energy, to maximize the signal-to-noise ratio in order to determine which phases can form on them.

## 3. Results and Discussion

### 3.1. General Framework

Radiolysis is the dissociation of molecules as a direct consequence of ionizing radiation, such as alpha particles, $\gamma$ or X-rays, or electrons, or indirectly by free high-energy neutrons. This general concept applies to water, too. After being exposed to radiation, water undergoes a breakdown sequence, and in a mass of liquid, some chemically aggressive species form. Figure 1 shows a schematic representation of the processes occurring at different moments and the species that form when water molecules are subjected to radiolysis [12,13].

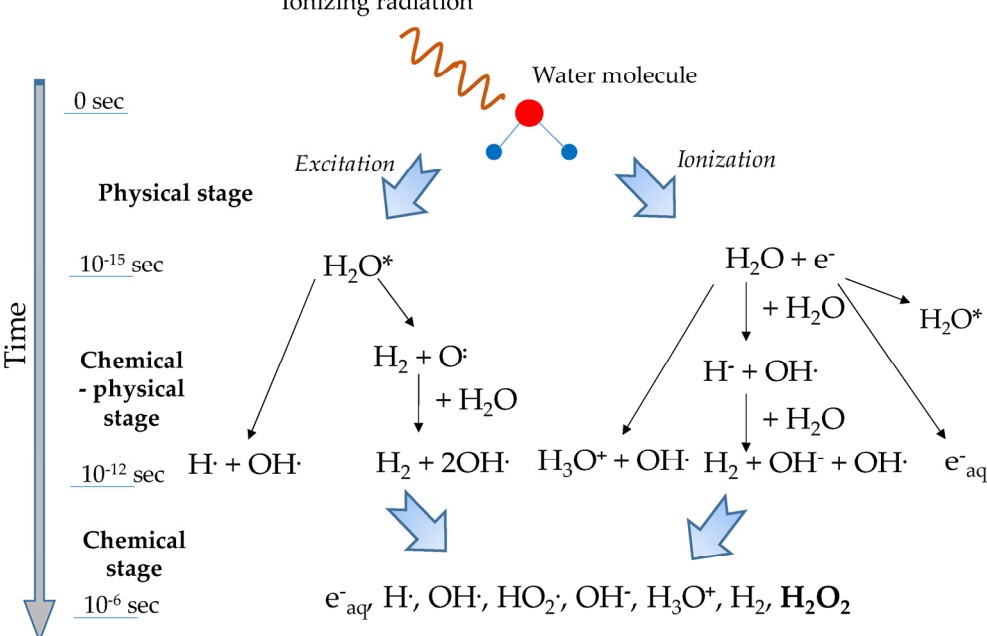

**Figure 1.** Chemical-physical reactions occurring during water radiolysis (inspired by references [12,13]).

Many of the species formed during this process have very high electrochemical potential and are strong oxidants, much stronger than substances such as ozone or chlorine. However, their half-lives are short, and therefore, they cannot accumulate to high levels, remaining at a steady state at low concentrations. Hydrogen peroxide is one of the final products, together with molecular hydrogen, and thus its concentration can increase over time [15,16,19].

Water radiolysis can produce both highly oxidizing and reducing species, and hydrogen peroxide can act both as an oxidizer and as a reductant, depending on the pH [20]. Thus, the specific general redox trend may be dominated by specific conditions such as temperature, the presence of impurities, and similar parameters. However, iron-based alloys are virtually stable in alkaline environments; thus, it seems more interesting to investigate their behavior in acid media, especially considering that water in industrial plants often has a pH lower than 7, because of the unavoidable presence of $CO_2$ in solution.

$H_2O_2$ can easily decompose on various surfaces, and this phenomenon is catalyzed by metals; therefore, in the case of steels and other alloys, the redox conditions are established by the interactions between reactive radiolysis products and metallic surfaces and the corrosion products formed by these reactions.

A few metals, like molybdenum and tungsten, react directly with hydrogen peroxide, while pure iron can undergo superficial alterations. However, steels are generally resistant to corrosion, unless the pH drops below 7. The semi-reaction for hydrogen peroxide under acidic pH is [21]:

$$H_2O_2 + 2H^+ + 2e^- \rightarrow 2H_2O, \; E_0 = 1.763 \text{ V}$$

so it can oxidize many metals to their highest valence states. Often ions corresponding to these states are more soluble in aqueous media, and this can explain why there is a release from the metals to the solution. The environment becomes more aggressive as the pH level decreases.

Iron can be oxidized initially to $Fe^{2+}$, and it, in turn, to $Fe^{3+}$, according to [22]:

$$Fe^{2+} + 2e^- \rightarrow Fe, E_0 = 0.44 \text{ V}$$

$$Fe^{3+} + e^- \rightarrow Fe^{2+} E_0 = 0.771 \text{ V}$$

The potential of hydrogen peroxide is higher than the potentials associated with both semi-reactions of iron. This implies that oxidation can indeed occur [8,23,24]. Figure 2 draws a diagram of the process.

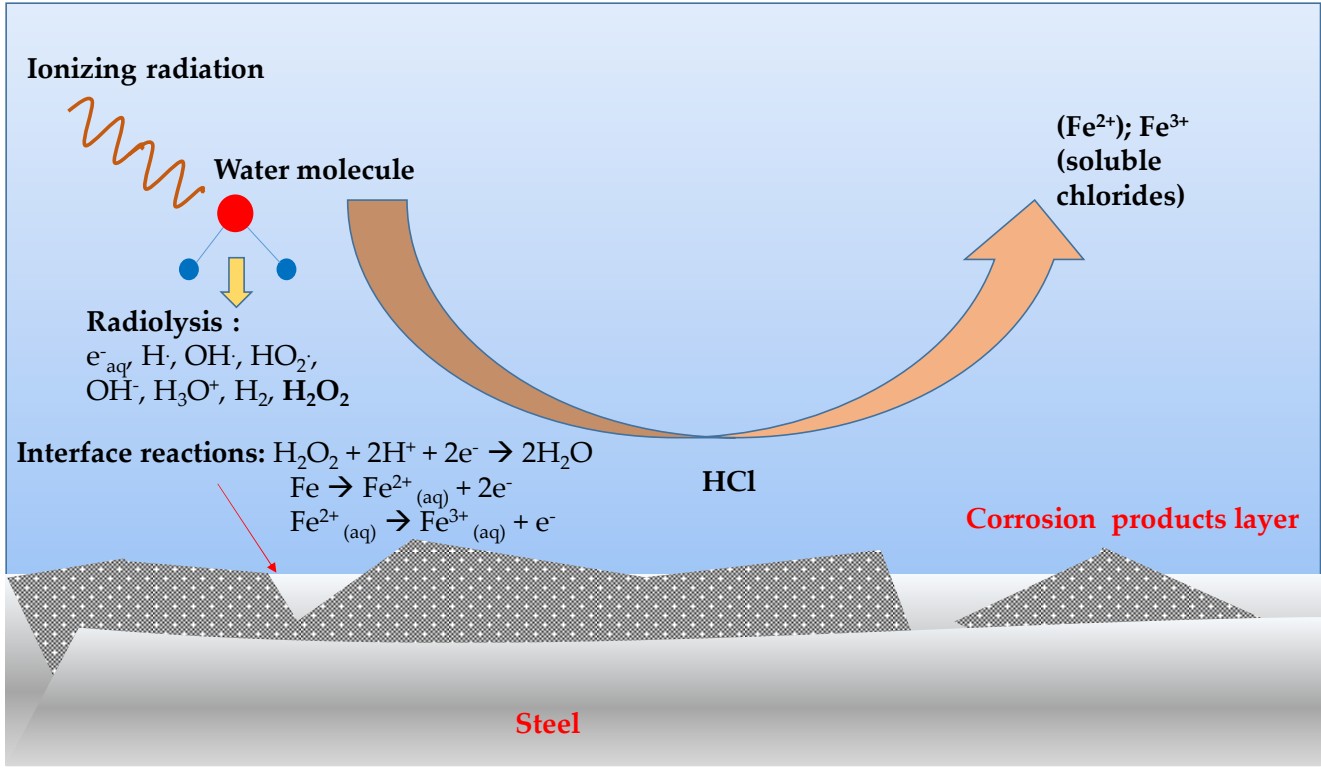

**Figure 2.** Schematic representation of steel corrosion due to radiolysis process.

The chemical paths of this process strongly depend on the composition and structure of the steel [25,26] and the product layer that forms on the surfaces. In general, corrosion also produces soluble metallic species and, in the presence of chloride anions, the final product is the soluble Fe(III) chloride. In these oxidative conditions, weight loss is expected.

*3.2. Corrosion Experiments*

Steel corrosion under the experimental conditions used in this work is associated with a very small, but still measurable, sample mass loss according to the electro-chemical mechanism described previously.

Figure 3 shows the mass variation of these alloys, $\Delta P$ %, expressed as $(P_0 - P_t)/P_0 \times 100$ ($P_0$ and $P_t$ being the initial weight and the weight of the metals at the moment t), as a function of time, for samples kept for different times at 70 °C under the action of a solution of hydrogen peroxide at 35% and with the addition of 1%, 5%, and 10% in volume of 1 M HCl. In the spirit of clarity, the behavior of different steels is shown in separate figures.

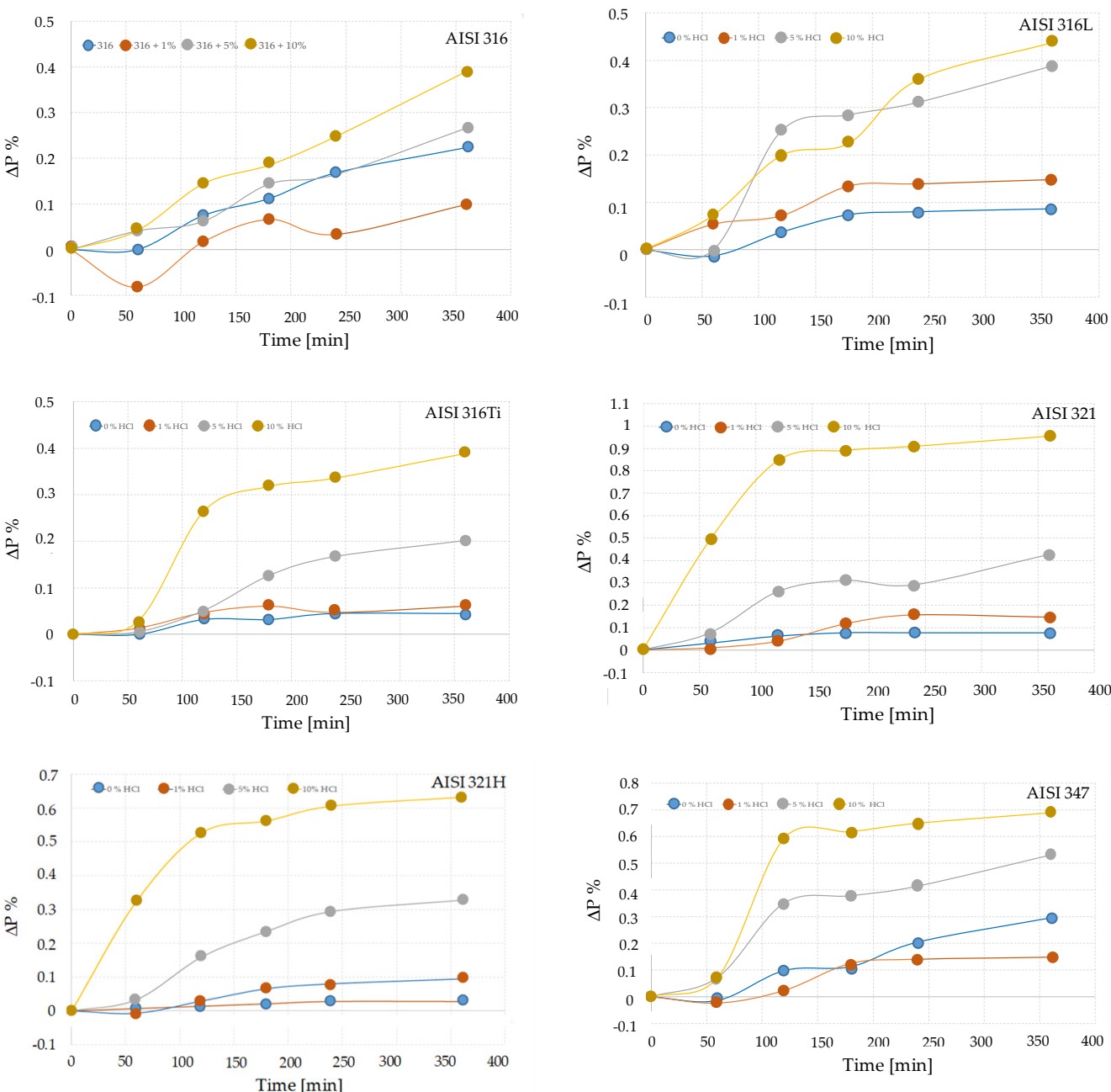

**Figure 3.** Mass variation for different stainless steels exposed to $H_2O_2$ 35% at 70 °C and different HC additions.

It can be observed that the mass of the samples decreases over time. There may be fluctuations in the general trend of mass loss and, in some cases, the variation is drastic from one stage to another, but in most of the cases, this can be attributed to the detachment of tiny portions from the individual sample rather than to actual chemical-physical behavior. The samples with higher HCl content show a faster decrease, except for a few samples when treated with low pH. Here, a small increase occurs in the initial stage of the process, which lasts for less than 30 min. This weight gain, which corresponds to a negative ∆P (according to the definition used here) is quite surprising. The first aspect that must be kept in mind is that these weight variations are appreciable in relative terms, but correspond to very small absolute changes (of the order of a few ten thousandths of a gram or less), very close to the effective sensibility and resolution of a normal balance, and for this, the relative experimental error is proportionally greater. The general trend is towards a continuous

decrease in weight and consistent with the solutions used, but the measured curves may have a non-monotonic trend for this reason. However, an explanation can also be sought in the possible formation of a thin layer of oxides and other iron compounds, which dissolves over longer times and leads to the actual weight loss that is measured. In fact, the color of most of the samples changes at the beginning, becoming weakly brownish, and this indicates that a chemical reaction on the surface occurs. The color of the solution turns into pale yellow, especially at higher pH and long times. This additional layer may be likely extremely thin and widespread over the entire surface of the sample, making it difficult to identify.

This could also offer an explanation for the fact that for certain samples, the $C_R$ is higher using pure hydrogen peroxide than using hydrogen peroxide with the addition of 1% HCl, which may be equally surprising. The mass loss results from two distinct phenomena—the oxidation of iron and the solubilization of oxidized compounds in the aqueous environment—and, likely, the former is a faster phenomenon than the dissolution of corrosion products. The hydrogen peroxide, being very concentrated, acts as a strong oxidant, even without adding acids, but the process proceeds in stages and initially, the metallic iron is oxidized to $Fe^{2+}$, which is later oxidized to $Fe^{3+}$, in water solution [27]. An electrochemical process takes place and the surface of steel acts as an anode where atoms of iron are oxidized and released in solution. Upon decreasing the pH, i.e., increasing the HCl content, the electrochemical potential of hydrogen peroxide becomes even higher, and oxidation to $Fe^{3+}$ may occur directly on the surface of samples. However, according to the Pourbaix diagram of iron [28], trivalent iron oxide, in the form of $Fe_2O_3*nH_2O$, is stable up to relatively low pH and does not dissolve immediately. The formation of a small amount of this compound is associated with an increase in the weight of the sample rather than its decrease. Further HCl additions drive the system far away from the stability field for this oxide, which, for this reason, dissolves more easily. In these cases, the samples monotonically lose weight, in a manner more regular and linked to the quantity of hydrochloric acid added.

A second consideration is that, after about two hours, even if the samples continue to lose mass, this occurs at a much slower rate, and two regimes can be clearly identified. A reason for this finding may be the self-decomposition of hydrogen peroxide, which, especially at high temperatures, is a rather rapid competing phenomenon, often catalyzed by the metal surfaces themselves. For this reason, the $C_R$, expressed as mass loss per minute, is estimated based on the mass variation measured after two hours.

Figure 4 shows the corrosion rate for different types of steel, subjected to different types of acidic environments. The insert shows the relative corrosion speed of each material compared to that of hydrogen peroxide alone. It is worth noting that concentrated hydrogen peroxide solutions are themselves acidic, although weakly [29], meaning that a mass loss occurs, athough small, even in the absence of HCl.

It must be remembered that this value of the corrosion rate is for these specific conditions. By changing some parameters, for instance, the $H_2O_2$ concentration or HCl content, a different $C_R$ would be found. The corrosion rate, therefore, should be understood as a means of examining the process, especially from a microscopic and mechanistic point of view, rather than as a means of describing a macroscopic process. Nevertheless, all the curves have, in general, a very similar shape, i.e., for a specific material, the reponse to mild conditions and long periods is very similar to that in the case of more severe conditions with short times. It can be stated that the corrosion process goes through the same stages, regardless of the speed with which they occur. The use of drastic conditions simply accelerates the phenomenon, allowing the long-term behavior of corrosion to be investigated in a short time.

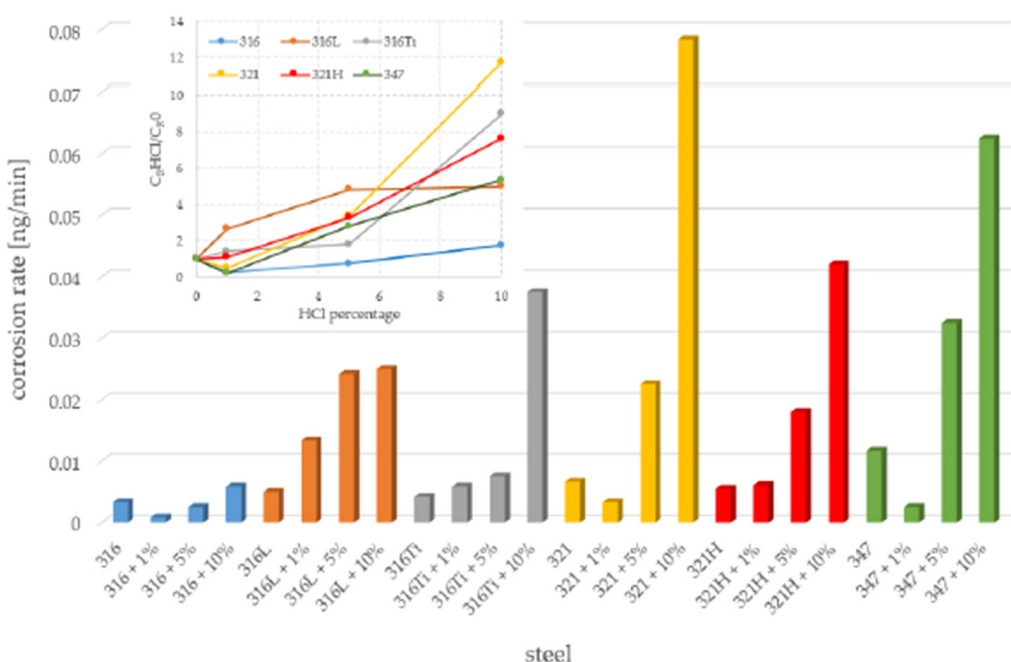

**Figure 4.** The corrosion rate for different types of steels using 1 M HCl. The insert shows the relative corrosion speed of each material compared to that of hydrogen peroxide alone.

This type of approach is often usefully used also in other fields of materials science, for example, the study of mechanical properties. For instance, the time–temperature superposition principle is an important concept for the polymer science and for the science of glasses and high-viscosity liquids [30,31]. It is used to determine temperature-dependent mechanical properties such as the elastic modulus or viscoelastic behavior from the known values at a reference temperature. The elastic relaxation modulus under a periodic excitation for a viscolestatic material depends on the temperature and on the frequency. Under certain hypotheses and assumptions [32], it is possible to determine the response of the material in a certain frequency and temperature range, from data known in other ranges. This allows extrapolation of these properties much beyond the field of experimental frequencies, which typically range from 0.01 to 100 Hz.

Therefore, as a first approximation, it seems reasonable to imagine that the response to mild corrosive processes at very long times can be modeled by studying the behavior at short times in more extreme conditions. The investigation of the steel responses under aggressive conditions can be an efficient means of studying the behavior over long periods, which is more difficult to carry out experimentally.

For a given material, the immediate corrosion rate rises as the hydrochloric acid 1 M content increases from 1% to 10%. However, it is crucial to note that this rate is significantly influenced by the nature of the starting material. In other words, the behavior is contingent upon the specific type of steel. Only after a certain period, again depending on the type of steel, mass loss increases continuously. Despite this, Raman spectra did not show significant differences between samples over time. In general, the Raman signal, like that of most metallic systems and alloys, was weak and had no particular spectral features. If the surface layer of oxides or compounds became thick over time, the appearance of some characteristic peaks or bands, which would allow the identification of the phases present, should occur, but this was not observed.

Oxidation from Fe to $Fe^{2+}$ and/or $Fe^{3+}$ and dissolution are not processes that should be imagined as consecutive, but rather as parallel and that they proceed with continuity.

Hydrochloric acid has the primary effect of increasing the oxidative capacity of the peroxide, leading to the formation of soluble chlorides [9,17,22], while, under weak acidic-to-basic conditions, oxidation of iron leads to the formation of insoluble phases, mostly $Fe(OH)_2$, $Fe_3O_4$, and $Fe_2O_3$, according to the Pourbaix diagram [33,34].

Nevertheless, the corrosion resistance of steel is contingent upon various factors, predominantly its chemical composition and the presence of specific secondary elements intentionally added to enhance it. For instance, steels with some titanium added are less prone to corrosion under these conditions.

Additionally, the microstructure—whether it is austenitic, ferritic, martensitic, or of another type—and the treatments the steel has undergone significantly contribute to its corrosion resistance.

The metallurgical structure is defined by the composition, i.e., in particular, by the balance of the elements (Mo, Si, Ti, Nb, Ta, Al) which stabilize the ferrite by widening its field of existence and those which, instead, stabilize the austenite (Ni, Co, N, Mn, Cu, as well as C). The relationship between the structure of steel and its composition is described by the Schaeffler diagram [2,7,35].

The steels considered in this work, because of their composition, are of the austenitic type. The crystalline structure of these steels is austenite (face-centered cubic) and in general, they have excellent corrosion resistance, good weldability, high strength at elevated temperatures, and improved machinability [36].

For these reasons, it would be more meaningful to compare steels within the same family. But even so, it seems there are some general rules that all steels follow, allowing us to generalize some considerations.

It is widely recognized that the corrosion resistance of stainless steels, under standard conditions, is attributed to the formation of a thin, self-healing, passivating film on their surface, facilitated by the presence of chromium [14,15,18]. It is important to note that for a specific group of steels, the $C_R$ is lower when the chromium content is high, i.e., the protective property remains even under strongly oxidative conditions. Similar observations can be done for nickel content. Nickel can increase corrosion resistance: it is an austenite stabilizer and widens the austenite region and contracts the ferrite region in steel. Nickel improves the resistance against the corrosion and oxidation at elevated temperatures. Nickel improves mechanical properties such as toughness and strength [37], by refining the grain size. Elemental nickel has a lesser diffusion coefficient compared to many elements and diffuses into steel crystalline lattice slowly [38].

However, regardless of the specific characteristics of the steels considered, it can be stated that the protective effect of the main additives is not altered by forced oxidation. Therefore, in general, the $C_R$ can be expected to be particularly high when the content of both elements is low. Although a greater number of compositions should be taken into consideration, in particular, samples in which, in turn, one of the two contents is low, it is possible to recognize, by plotting the corrosion rate as a function of both Cr and Ni contents, that the corrosion peaks are in the expected areas. Figure 5 shows the $C_R$ and hours of reaction with $H_2O_2$ 35% plus 10% 1 M HCl at 70 °C, as a function of both Cr and Ni contents. It should be kept in mind that steels are functional materials and that their compositions are well established for the functions they are intended to perform. For instance, steel 316 is often used for pipe clamping. Thus, compositions cannot go outside the specifications for that material type.

To investigate the corrosion/dissolution processes in more detail, some samples were treated by changing the experimental conditions, i.e., processing temperature, $H_2O_2$ concentration, or HCl concentration. In order to make corrosion faster and minimize experimental errors, the tests were conducted using concentrated (35% $w/v$) instead of diluted hydrochloric acid.

Figure 6A,B show, respectively, the effect of different concetrations of hydrogen peroxide on the mass variation and $C_R$ of samples of 316Ti steel measured after two hours of treatment at 70 °C, with the addition of 10% of HCl. The other steels have similar behaviors in these conditions, in the sense that the absolute values are different for each of them, but the trends are very similar.

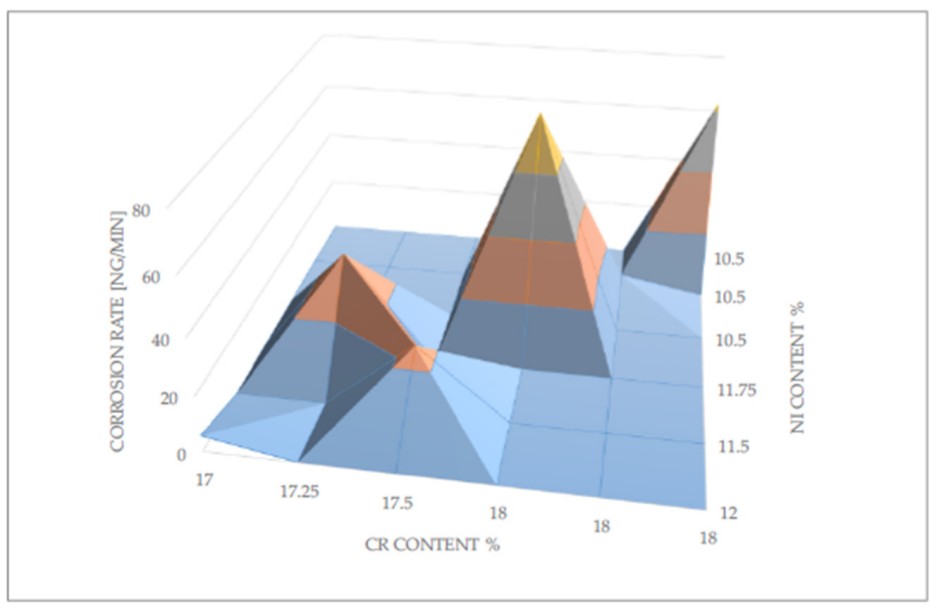

**Figure 5.** Corrosion rate as function of Cr and Ni contents measured after 2 h of reaction with $H_2O_2$ 35% and 10% 1 M HCl at 70 °C.

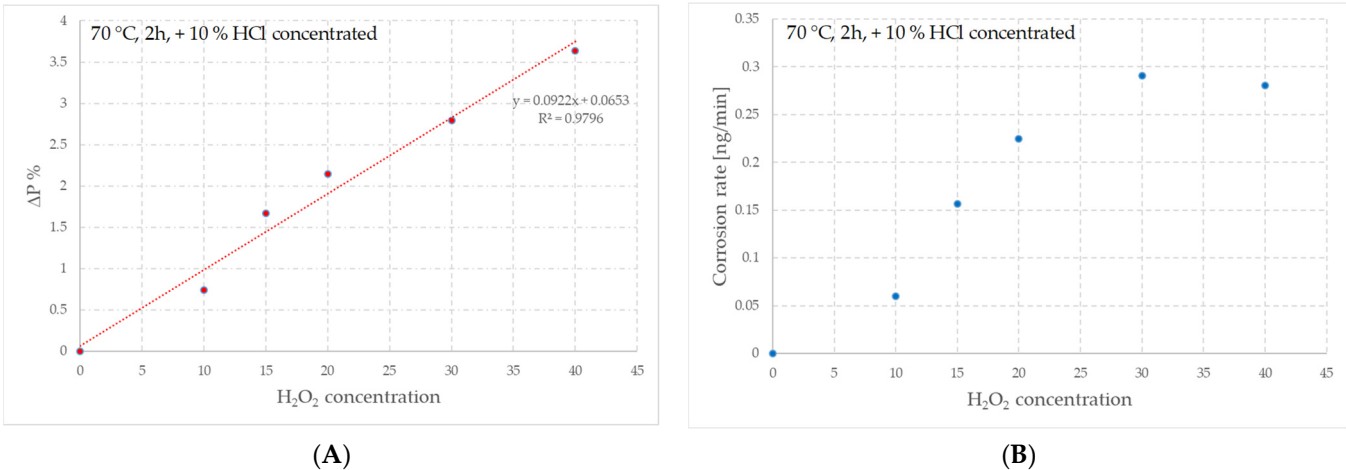

(**A**)                                                                                              (**B**)

**Figure 6.** Dependence of the mass loss (**A**) and the corrosion rate (**B**) as a function of $H_2O_2$ concentration.

The mass decreases linearly with peroxide concentration, whereas for $C_R$, this is only true up to 30%, because when the concentration is equal to 40%, the dissolution is a little slower than expected. It is possible that in these conditions, the self-decomposition of hydrogen peroxide is extremely rapid.

Figure 7A,B show, for the same steel, the mass variation and the $C_R$ trend as functions of the temperature, keeping constant the time of treatment, 2 h; the $H_2O_2$ concentration, 35%; and the addition of concentrated HCl, 10%.

In this case, the mass loss dependence is better fitted by an exponential curve, rather than a linear one. In fact, corrosion at low temperatures is very modest, while it is much more rapid at high temperatures. This is well compatible with a thermally activated process: corrosion/dissolution kinetics is favored by increasing temperature.

These observations highlight the role of peroxide in the oxidation process. In general, redox reactions are interpreted on the basis of multistep kinetic models which postulate the formation and subsequent fast reactions of reactive intermediates. However, under these experimental conditions, at the liquid/steel interface, the concentrations of iron ions can be considered stationary and constant, because they derive from an effectively infinite source. This mean that, from a kinetic point of view, the process is mainly controlled by

the hydrogen peroxide and actually, the reaction becomes faster as its concentration and temperature increase. From a microscopic point of view, the corrosion process is controlled by the reactions taking place at the surface of steel pieces, regardless of the microstructural evolution of corrosion in the material and its chemical composition.

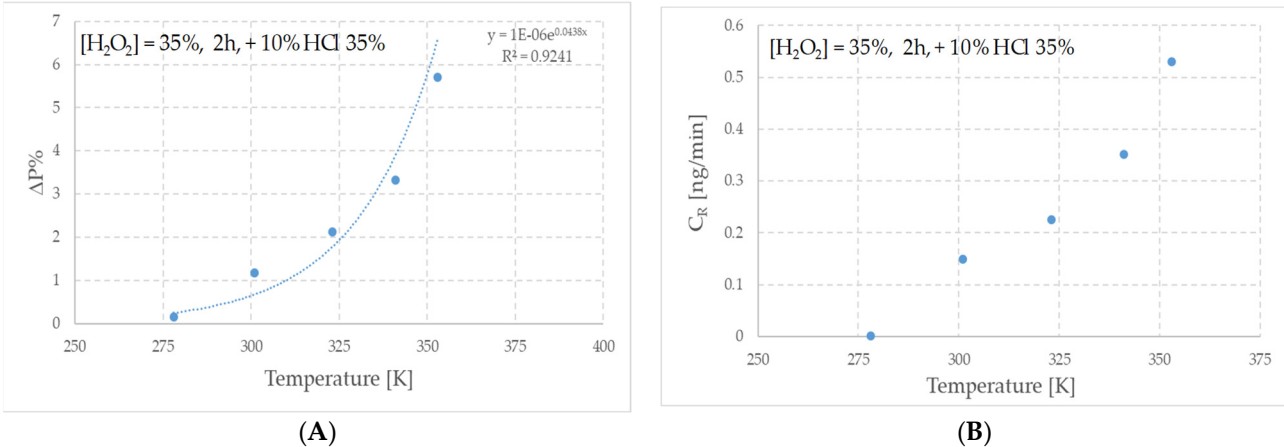

**Figure 7.** Dependence of the mass loss (**A**) and the corrosion rate (**B**) as a function of temperature.

This is very important when it is possible that hydrogen peroxide or possibly other oxidizing agents can accumulate over time and when the operating temperature is high and must be taken into account when designing materials and evaluating the lifespan of a steel.

### 3.3. Microscopic Characterization

Figure 8 shows the SEM images of the surfaces of AISI 321H and AISI 347 samples, treated at 70 °C with 10% of 1 M HCl added, at the beginning of the process (t = 0) and their evolution as a function of the time, i.e., after two hours of treatment. They can be considered typical examples of the behavior of steels in these conditions. It should be noted that the morphology of the porous surfaces of steels presents characteristics that remain the same at each stage of the process, as a function of time, or for more or less mild or aggressive conditions, but the number and abundance of corrosion marks changes, and they may eventually evolve, but with continuity. This means that the evolution of corrosion of a specific steel follows a specific path through stages that depends on the nature of that steel, and that the external conditions determine the kinetics, so that a steel subjected to mild conditions but for a short time is very similar to one that has been subjected to more aggressive conditions, but for a short time.

In other words, hydrogen peroxide and pH do not determine the evolution of the corrosion of a steel and the aspect of a corroded sample and its morphology, but their concentration and value are the driving forces of the initial stages of the oxidative process.

The appearance of the treated surfaces depends, under the same experimental conditions, first of all on the type of the steel, although other factors such as possible mechanical stress, the history of the sample, fatigue, grain size and others [39] can play an important role. In the early stages, it is difficult to identify specific areas of corrosion on the surfaces. On the contrary, after some time, there are signs of very localized corrosion, excluding that this oxidation/dissolution process is a type of general corrosion that is a homogenous and continuos loss of atoms from the whole surfaces. In general, after a sufficiently long time, some deep marks appear on them, and in most cases, some large craters form. They do not have a constant size in all areas of the samples, but, on the contrary, different widths and depths, and often the wider ones are proportionately much deeper than the smaller ones. This indicates that they are not all formed at the same time, but randomly throughout the treatment period and the whole surface, so that the first ones that appear become larger and the last ones have little time to grow. These marks, especially the larger ones, are often

separated by thin walls that meet at angles of about 120°. The final aspect of the corroded surfaces of all the samples is somehow similar, that being the presence of many holes and craters, but their origin in different steels is different. In some cases, as for steel 347, it appears that these large holes are the final stage of the pitting process. Instead, in other cases, like steel 321, it seems that some grains have detached from the surface due to strong corrosion that has dug around them.

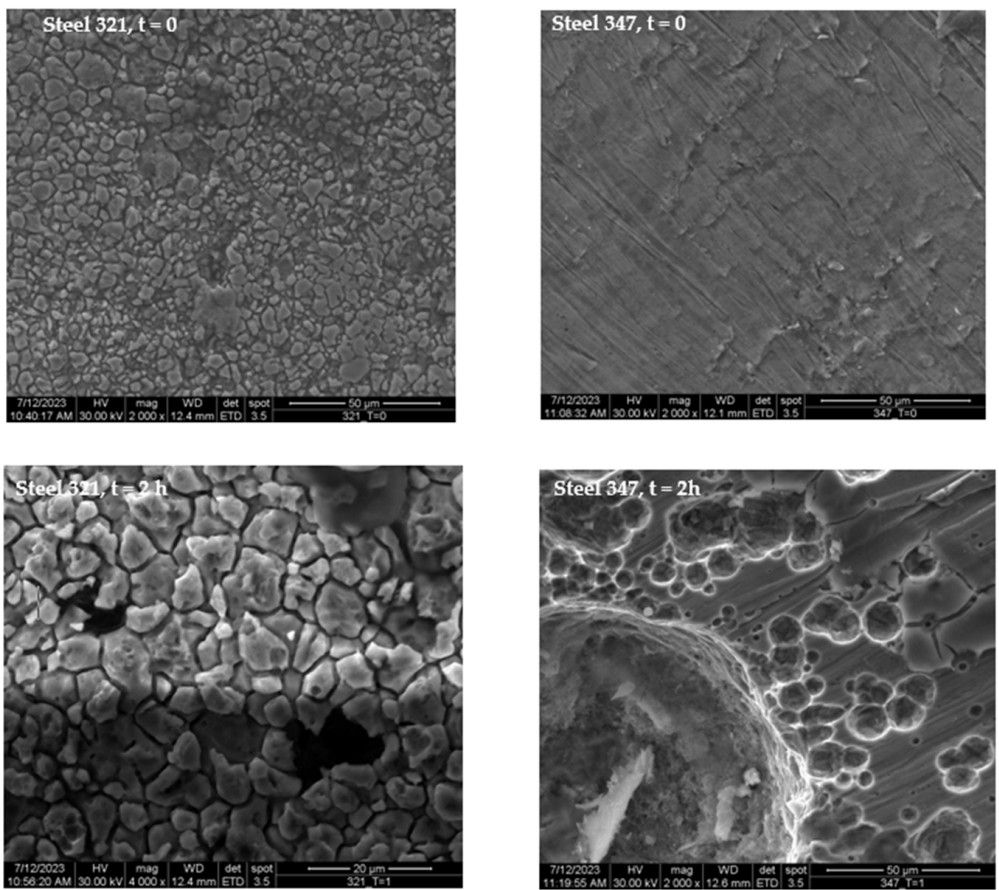

**Figure 8.** SEM images of AISI 321H (**left column**) and AISI 347 (**right column**) at t = 0 (**upper line**) and after 2 h (**lower line**), at 70 °C and H$_2$O$_2$ 35% plus 10% of 1 M HCl.

In general, a corrosion process begins in areas where the surface energy is the highest [39,40], because of the presence of impurities, local stresses, or other factors, and the driving force is the difference between their electrochemical potentials and the cores of grains themselves. Once started, corrosion preferentially continues around those areas, and it is much faster there than where it has not yet begun. In other words, corrosion can proceed much more easily if the passivating film is no longer compact. If the high-energy zones correspond to the grain boundaries, erosion predominantly takes place there, manifesting as intergranular corrosion, a form of relatively rapid and localized corrosion associated with a defective microstructure known as carbide precipitation [41]. In this process, the crystal boundaries of the material are more susceptible to corrosion compared to their interiors. Grain boundaries are intrinsically disordered areas, with a lot of crystallographic stresses, and are rich in secondary phases. Often among these secondary phases, chromium carbide is often found, and its presence induces a local decrease in the average chromium content. Formation of these carbide particles in the grain boundaries depletes the surrounding metal of chromium and reduces its corrosion resistance, allowing the steel to corrode preferentially along the grain boundaries, and they become more and more marked, leading eventually to the detachment of some grains [42].

A steel of this type subjected to this type of environment evolves over time, and intergranular corrosion slowly leads to the detachment of the grains, with a rate which depends on aggressiveness of the environment. Figure 9, in fact, shows back-scattered SEM images of the surface of a 321H steel treated at 70 °C with a solution of 35% $H_2O_2$ and 10% concentrated HCl, after only 15 min.

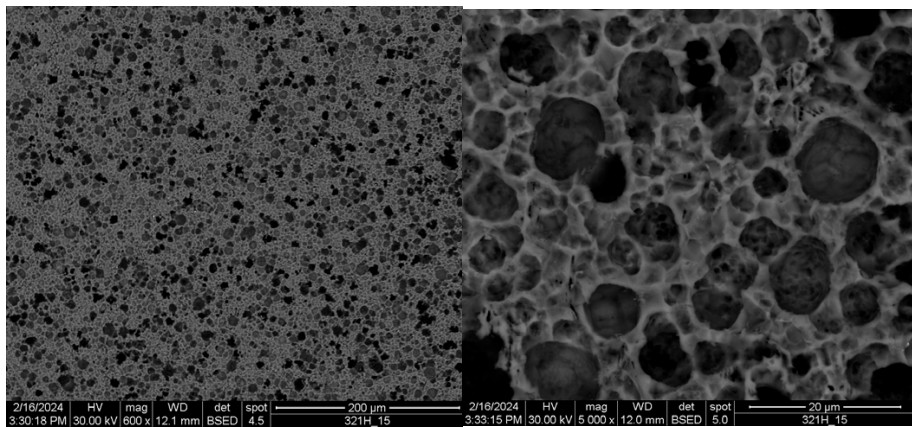

**Figure 9.** Back-scattered SEM images of an AISI 321H sample treated for 15 min at 70 °C and $H_2O_2$ 35% plus 10% of concentrated HCl.

However, in some steels, small quantities of secondary elements are added—niobium and tantalum, in the case of 347 steel—precisely to avoid the precipitation of chromium carbide [43]. In this case, the grain boundaries are less prone to be corroded; therefore, there are no preferential areas where corrosion can start. For these steels, SEM analysis has shown that the grain boundaries do not appear more corroded than the rest of the surface even after a long contact time with the oxidizing solution. On the contrary, corrosion begins at random points on the surfaces, probably due to the presence of structural defects or impurities. For these materials, corrosion is very localized, like in the pitting process, and leads to the creation of small holes in metal. The driving force is, in this case, the de-passivation of a small area. Even the mechanical action of the microbubbles of oxygen and water vapor that form on surfaces due to the decomposition of hydrogen peroxide may play an important role in this. Once the protective layer is disrupted, the corrosive process can start, and it can even proceed in an autocatalytic manner, thanks to local electrochemical processes. A schematic depiction of these processes is shown in Figure 10.

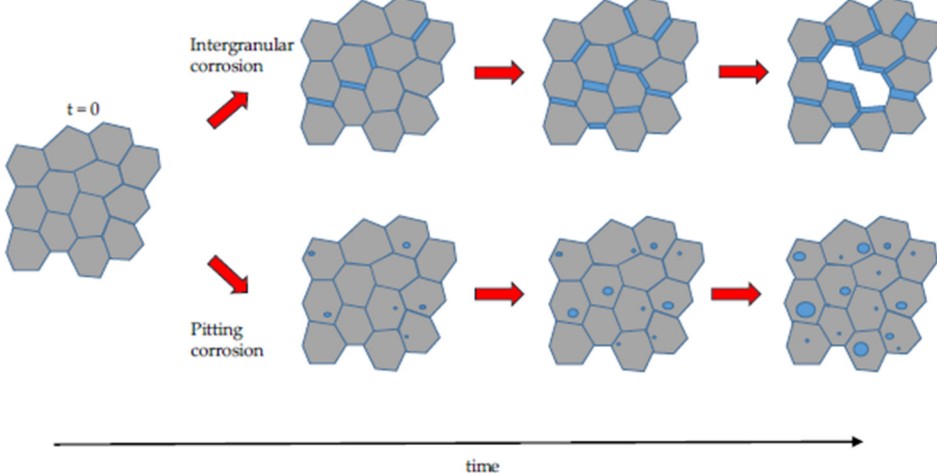

**Figure 10.** Schematic representation of the evolution of corrosion of different types of steels, prone to intergranular corrosion or pitting, subject to highly aggressive conditions.



The two models described, in any case, should be imagined as two limiting and ideal situations: the behavior of steel is, rather, a combination of the two, in which, depending on the material considered and its chemical composition, one is prevalent. The effects after very long times tend to be very similar.

## 4. Conclusions

The study of corrosion of stainless steels is extremely important, especially when they are used in applications that subject them to very aggressive environments. Although stainless steels are resistant materials, in these cases, they can degrade much more quickly. This is the case of steels used in nuclear plants in contact with water. In this situation, the phenomenon of radiolysis, i.e., the decomposition of water molecules due to radiation, should not be neglected. It leads to the formation of very aggressive chemical species, and also to the formation of hydrogen peroxide. These species can, albeit slowly, induce severe corrosion in most steels. One way to study corrosion in an accelerated manner is to use much higher-than-normal concentrations of peroxide under acidic conditions, which makes it an even more energetic oxidizer. In this way, steels can be measurably corroded in a matter of hours. The corrosion path strongly depends on the nature and chemical composition of the steels, both from a kinetic and a microscopic morphology point of view. In a general way, the chromium and nickel which gives steels their corrosion resistance properties continue to function even in much harsher conditions, and the corrosion rate is low when their content is high. Corrosion can be of the intergranular type, and in this case, the final stage is the detachment of some grains from the samples or of the pitting type, i.e., with the formation of localized craters on the surface, which can grow over time. Whether a specific steel follows one or the other path depends on its chemical composition, and in particular, on the presence of certain secondary elements.

The idea of exploiting accelerated corrosion allows studying of the behavior of stainless steels on a much more accessible time scale. In principle, this strategy might also be applied to other types of steels and alloys.

**Author Contributions:** Conceptualization, S.C. and A.U.; methodology, A.U.; validation, A.R. and S.B.; investigation, A.U., C.T., S.C., A.G., G.M., F.G., G.P. and S.B.; writing—original draft preparation, A.U.; writing—review and editing, S.C. and A.U. All authors have read and agreed to the published version of the manuscript.

**Funding:** This research received no external funding.

**Institutional Review Board Statement:** This study does not require ethical approval.

**Data Availability Statement:** Data are contained within the article.

**Conflicts of Interest:** The authors declare no conflicts of interest.

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
