# Peer review of "A Study of Accelerated Corrosion of Stainless Steels under Highly Oxidizing Conditions"

_coatings, doi:10.3390/coatings14040390_

Round 1

Reviewer 1 Report

Comments and Suggestions for Authors

1、Figure 1 needs to be cited

2、The flowchart in Figure 2 is too simple.

3、Based on the results in Figure 3, a detailed analysis should be conducted on the weight loss patterns and causes of different samples at different HCL concentrations;

4、There are multiple incomplete sentences, unclear sentences, and grammatical errors in the text.

5、The discussion did not properly combine the information in all figures, and figures and the explanations are detached.

6、The curves figures in the text are not clear.

Comments on the Quality of English Language

English language needs to be improved. 

Author Response

Dear reviewer

“1、Figure 1 needs to be cited

2The flowchart in Figure 2 is too simple.

3Based on the results in Figure 3, a detailed analysis should be conducted on the weight loss patterns and causes of different samples at different HCL concentrations;

4There are multiple incomplete sentences, unclear sentences, and grammatical errors in the text.

5The discussion did not properly combine the information in all figures, and figures and the explanations are detached.

6The curves figures in the text are not clear.

All the comments you made are very relevant and we have tried to respond to them as best we can. We have corrected all the typos and other errors. We have also added references where required. The quality of the figures has been improved and in particular figure 2 has been completed to provide the useful information derived from it.

Regarding point three, we have added a discussion supported by some new references, regarding the behavior of steels. You are right that an analysis with further acid additions would be useful, but we believe that the general behavior of these materials can be described sufficiently well even so. The oxidation of a ferrous alloy is a complex electrochemical process which realistically proceeds in stages. Experimental conditions, such as the amount of acid added, determine these stages and determine the change in weight over time. For example, the oxidation of iron leads to the formation of Fe2O3, which however is stable up to relatively low pH. Therefore, in some samples an increase in weight can initially be observed instead of a simple decrease. When acid is added in larger quantities, this oxide also dissolves easily.

We have tried to deepen the discussion of this point in the manuscript, also adding references and we hope that now the discussion is clearer.

Best regards

Alberto Ubaldini and all the authors.

Reviewer 2 Report

Comments and Suggestions for Authors

1.  Increase the font size of the text (including the axis title) within each figure and make it clear. It's very difficult to figure out.   

2. in Figure 3: What does it mean that ΔP% is negative?

3. in Figure 3: In the case of 'AISI 316', 'AISI321H', and 'AISI347' samples, unlike other samples (AISI 316L, AISI316Ti, AISI321), '316' and '0% HCl' show a larger ΔP% than '1~10% HCl'. why?

4. Change ‘concetrated’ to ‘concentrated’.

5. Throughout the manuscript: Please check again for typos and awkward phrases.

---------------------------------------------The end------------------------------------------

Comments on the Quality of English Language

Throughout the manuscript: Please check again for typos and awkward phrases.

Author Response

Dear reviewer

1. .  Increase the font size of the text (including the axis title) within each figure and make it clear. It's very difficult to figure out.   

  1. in Figure 3: What does it mean that ΔP% is negative?
  2. in Figure 3: In the case of 'AISI 316', 'AISI321H', and 'AISI347' samples, unlike other samples (AISI 316L, AISI316Ti, AISI321), '316' and '0% HCl' show a larger ΔP% than '1~10% HCl'. why?
  3. Change ‘concetrated’ to ‘concentrated’.
  4. Throughout the manuscript: Please check again for typos and awkward phrases

We have carefully read your comments, which are useful and very relevant.

We have tried to improve the quality of the figures and corrected all the errors that we missed in the first version.

Regarding your questions about the negative weight change, we believe that this could be the explanation. The corrosion process is certainly complex and it must be seen in an electrochemical framework. Iron oxidizes first to divalent Fe ion and then to trivalent Fe ion.

Concentrated hydrogen peroxide is easily able to take the first step and the surface of the steel releases ions into solution, where they are further oxidized. It is possible that the kinetics of the two processes are different. The electrochemical power of the peroxide increases by reducing the pH and the iron is oxidized to the trivalent state more easily and rapidly. However, at pH that is not too acidic (i.e. with the addition of only 1% hydrochloric acid) the iron oxide, Fe2O3 is still stable. Therefore, small amounts of it can accumulate on some samples and in this case, the weight initally increases instead of decreasing. Over time, also it dissolves. In other words, the weight variation is initially negative and for the same reason some steels have a higher resistance to corrosion when a little quantity of acid is added in respect to pure hydrogen peroxide. In the manuscript, we tried to better argue this idea by also adding some references.

Best regards

Alberto Ubaldini and all the authors.

Reviewer 3 Report

Comments and Suggestions for Authors

It is a nice contribution about stainless steel.

The influence of kinetics (temperature) can be regarded in more detail - but I am fine as it is. Some minor language editing may be usefull (better write nuclear power plant, page 3 line 119: change m7/v, figue 1: exposition to water).

Some figures should be improved (Quality Fig. 3,  Fig. 3 set all on same scale). 

Maybe you can also regard the relevance of the study for implant material, as implant material like Vitallium is also exposed to harsh conditions. Also mention the aspects for stainless steel rolling bearings (food industry / aeroplanes).

Author Response

Dear reviewer

“It is a nice contribution about stainless steel.

The influence of kinetics (temperature) can be regarded in more detail - but I am fine as it is. Some minor language editing may be usefull (better write nuclear power plant, page 3 line 119: change m7/v, figue 1: exposition to water).

Some figures should be improved (Quality Fig. 3,  Fig. 3 set all on same scale). 

Maybe you can also regard the relevance of the study for implant material, as implant material like Vitallium is also exposed to harsh conditions. Also mention the aspects for stainless steel rolling bearings (food industry / aeroplanes)”

Thank you very much for appreciating the idea behind our manuscript.

We apologize for all the typos in our manuscript. We very much agree with you that probably many of the observations made can be more general and applicable to other steels and alloys. There are certainly many other cases where steels are subjected to difficult or even extreme conditions. Among them, certainly steels that must work in turbines, at high temperatures, or subjected to great mechanical stress or the action of many chemical substances. Perhaps our idea could be useful for studying these cases too.

In the text, we have in fact talked about it now. However, appropriate experimental work would be needed to be sure.

Best regards

Alberto Ubaldini and all the authors.

Round 2

Reviewer 1 Report

Comments and Suggestions for Authors

The mancript can be accepted in present form.

Comments on the Quality of English Language

English language does not require revision